# Individualized Dosing Dynamics
# via Neural Eigen Decomposition

**Stav Belogolovsky**[*]
stav.belo@gmail.com

**Ido Greenberg**[*]
gido@campus.technion.ac.il

**Danny Eytan**[†]
biliary.colic@gmail.com

**Shie Mannor**[*‡]
shie@ee.technion.ac.il

## Abstract

Dosing models often use differential equations to model biological dynamics. Neural differential equations in particular can learn to predict the derivative of a process, which permits predictions at irregular points of time. However, this temporal flexibility often comes with a high sensitivity to noise, whereas medical problems often present high noise and limited data. Moreover, medical dosing models must generalize reliably over individual patients and changing treatment policies. To address these challenges, we introduce the Neural Eigen Stochastic Differential Equation algorithm (**NESDE**). NESDE provides individualized modeling (using a hypernetwork over patient-level parameters); generalization to new treatment policies (using decoupled control); tunable expressiveness according to the noise level (using piecewise linearity); and fast, continuous, closed-form prediction (using spectral representation). We demonstrate the robustness of NESDE in both synthetic and real medical problems, and use the learned dynamics to publish simulated medical gym environments.

## 1 Introduction

Sequential forecasting in irregular points of time is required in many real-world problems, such as modeling dosing dynamics of various medicines (pharmacodynamics). Consider a patient whose physiological or biochemical state requires continuous monitoring, while blood tests are only available with a limited frequency. Pharmacodynamics models often rely on an ordinary differential equation models (ODE) for forecasting. Additional expressiveness can be obtained via customized learned models, such as neural-ODE, which learns to predict the derivative of the process [Chen et al., 2018, Liu et al., 2019]. By predicting the *derivative*, neural-ODE can make irregular predictions at flexible time-steps, unlike regular models that operate in constant time-steps (e.g., Kalman filter, Kalman [1960] and recurrent neural networks, Rumelhart et al. [1986]).

However, real-world forecasting remains a challenge for several reasons. First, the variation between patients often requires personalized modeling. Second, neural-ODE methods are often data-hungry: they aggregate numerous derivatives provided by a non-linear neural network, which is often sensitive to noise. Training over a large dataset may stabilize the predictions, but data is often limited. Third, most neural-ODE methods only provide a point-estimate, while uncertainty estimation is often critical in medical settings. Fourth, for every single prediction, the neural-ODE runs a numeric ODE solver,

---

[*]Department of Electrical and Computer Engineering, Technion—Israel Institute of Technology, Haifa, Israel
[†]Department of Physiology and Biophysics, Faculty of Medicine, Technion—Israel Institute of Technology, Haifa, Israel
[‡]Nvidia Research

37th Conference on Neural Information Processing Systems (NeurIPS 2023).

Table 1: A summary of the features of NESDE. In Sections 5,6, NESDE is tested on synthetic and real medical data. In addition, each component of NESDE is studied experimentally as specified.

| Challenge | Solution | Experimental evaluation |
|---|---|---|
| Individualized modeling | Hyper-network with high-level features input | Hyper-network ablation (Appendix E.1) |
| Sample efficiency | Regularized dynamics: piecewise-linear with complex eigenvalues | Varying train size (Section 5, Appendix E.2); varying sparsity (Appendix E.3) |
| Uncertainty estimation | Probabilistic Kalman filtering | NLL evaluation (Sections 5,6) |
| Fast continuous inference | Spectral representation with closed-form solution | Time flexibility (Appendix E.4); interpretability (Appendix E.5) |
| Control generalization | Decoupling control from other inputs | Out of distribution control (Section 5, Appendix E.6) |

along with multiple neural network calculations of the derivative. This computational overhead in inference may limit latency-sensitive applications.

A fifth challenge comes from control. In the framework of retrospective forecasting , a control signal (drug dosage) is often considered part of the observation [De Brouwer et al., 2019]. However, this approach raises difficulties if the control is observed at different times or more frequently than other observations. If the control is part of the model output, it may also bias the train loss away from the true objective. Finally, by treating control and observations together, the patterns learned by the model may overfit the control policy used in the data – and generalize poorly to new policies.

Generalization to out-of-distribution control policies is essential when the predictive model supports decision-making, as the control policy may be affected by the model. Such decision-making is an important use-case of sequential prediction: model-based reinforcement learning and control problems require a reliable model [Moerland et al., 2020, Angermueller et al., 2019], in particular in risk-sensitive control [Yu et al., 2021, Greenberg et al., 2022, Greenberg and Mannor, 2021].

Section 4 introduces the Neural Eigen-SDE algorithm (**NESDE**) for continuous forecasting, which is designed to address the challenges listed above. NESDE uses a hypernetwork to provide an individualized stochastic differential equation (SDE), regularized to be piecewise-linear and represented in spectral form. The SDE derives a probabilistic model similar to Kalman filtering, which provides uncertainty estimation. Finally, the SDE decouples the control signal from other observations, to discourage the model from learning control patterns that may be violated under out-of-distribution control policies. Table 1 summarizes all these features.

Section 5 tests NESDE against both neural-ODE methods and recurrent neural networks. NESDE demonstrates robustness to both noise (by learning from little data) and out-of-distribution control policies. In Appendix E.5, the SDE model of NESDE is shown to enable potential domain knowledge and provide interpretability – via the predicted SDE eigenvalues. Appendix E.4 demonstrates the disadvantage of discrete methods in continuous forecasting.

In Section 6, NESDE demonstrates high prediction accuracy in two medical forecasting problems with noisy and irregular real-world data: (1) blood coagulation prediction given Heparin dosage, and (2) prediction of the Vancomycin (antibiotics) levels for patients who received it.

**Contribution:**

- We characterize the main challenges in continuous forecasting for medication dosing control.

- We design the novel Neural Eigen-SDE algorithm (NESDE), which addresses the challenges as summarized in Table 1 and demonstrated empirically in Sections 5 and 6.

- We use NESDE to improve modeling accuracy in two medication dosing processes. Based on the learned models, we simulate gym environments for future research of healthcare control.

## 1.1 Related Work

**Classic filtering:** Classic models for sequential prediction in time-series include ARIMA models [Moran and Whittle, 1951] and the Kalman filter (KF) [Kalman, 1960]. The KF provides probabilistic distributions and in particular uncertainty estimation. While the classic KF is limited to linear dynamics, many non-linear extensions have been suggested [Krishnan et al., 2015, Coskun et al., 2017, Revach et al., 2021, Greenberg et al., 2021]. However, such models are typically limited to a constant prediction horizon (time-step). Longer-horizon predictions are often made by applying the model recursively [Herrera et al., 2007, Bontempi et al., 2013], This poses a significant challenge to many optimization methods [Kolen and Kremer, 2001], as also demonstrated in Appendix E.4.

Limited types of irregularity can also be handled by KF with intermittent observations [Park and Sahai, 2011, Sinopoli et al., 2004] or periodical time-steps [Li et al., 2008].

**Recurrent neural networks:** Sequential prediction is often addressed via neural network models, relying on architectures such as RNN [Rumelhart et al., 1986], LSTM [Hochreiter and Schmidhuber, 1997] and transformers [Vaswani et al., 2017]. LSTM, for example, is a key component in many SOTA algorithms for non-linear sequential prediction [Neu et al., 2021]. LSTM can be extended to a filtering framework to alternately making predictions and processing observations, and even to provide uncertainty estimation [Gao et al., 2019]. However, these models are typically limited to constant time-steps, and thus suffer from the limitations discussed above.

**Neural-ODE models:** Parameterized ODE models can be optimized by propagating the gradients of a loss function through an ODE solver [Chen et al., 2018, Liu et al., 2019, Rubanova et al., 2019]. By predicting the process *derivative* and using an ODE solver in real-time, these methods can choose the effective time-steps flexibly. Uncertainty estimation can be added via process variance prediction [De Brouwer et al., 2019]. However, since neural-ODE methods learn a non-linear dynamics model, the ODE solver operates numerically and recursively on top of multiple neural network calculations. This affects running time, training difficulty and data efficiency as discussed above. While neural-ODE models have been studied for medical applications with irregular data [Lu et al., 2021], simpler models are commonly preferred in practice. For example, the effects of Heparin on blood coagulation is usually modeled by either using discrete models [Nemati et al., 2016] or manually based on domain knowledge [Delavenne et al., 2017].

Our method uses SDE with piecewise linear dynamics (note this is *different* from a piecewise linear process). The linear dynamics per time interval permit efficient and continuous closed-form forecasting of both mean and covariance. Schirmer et al. [2022] also rely on a linear ODE model, but only support operators with real-valued eigenvalues (which limits the modeling of periodic processes), and do not separate control signal from observations (which limits generalization to out-of-distribution control). Our piecewise linear architecture, tested below against alternative methods including De Brouwer et al. [2019] and Schirmer et al. [2022], is demonstrated to be more robust to noisy, sparse or small datasets, even under out-of-distribution control policies.

## 2 Preliminaries: Linear SDE

We consider a particular case of the general linear Stochastic Differential Equation (SDE):

$$dX(t) = [A \cdot X(t) + \tilde{u}(t)] + dW(t) \tag{1}$$

where $X : \mathbb{R} \to \mathbb{R}^n$ is a time-dependent state; $A \in \mathbb{R}^{n \times n}$ is a fixed dynamics operator; $\tilde{u} : \mathbb{R} \to \mathbb{R}^n$ is the control signal; and $dW : \mathbb{R} \to \mathbb{R}^n$ is a Brownian motion vector with covariance $Q \in \mathbb{R}^{n \times n}$.

General SDEs can be solved numerically using the first-order approximation $\Delta X(t) \approx \Delta t \cdot dX(t)$, or using more delicate approximations [Wang and Lin, 1998]. The linear SDE, however, and in particular Eq. (1), can be solved analytically [Herzog, 2013]:

$$X(t) = \Phi(t) \left( \Phi(t_0)^{-1} X(t_0) + \int_{t_0}^{t} \Phi(\tau)^{-1} \tilde{u}(\tau) d\tau + \int_{t_0}^{t} \Phi(\tau)^{-1} dW(\tau) \right) \tag{2}$$

where $X(t_0)$ is an initial condition, and $\Phi(t)$ is the eigenfunction of the system. More specifically, if $V$ is the matrix whose columns $\{v_i\}_{i=1}^{n}$ are the eigenvectors of $A$, and $\Lambda$ is the diagonal matrix

whose diagonal contains the corresponding eigenvalues $\lambda = \{\lambda_i\}_{i=1}^n$, then

$$\Phi(t) = V e^{\Lambda t} = \begin{pmatrix} | & & | & & | \\ v_1 \cdot e^{\lambda_1 t} & \cdots & v_i \cdot e^{\lambda_i t} & \cdots & v_n \cdot e^{\lambda_n t} \\ | & & | & & | \end{pmatrix} \tag{3}$$

If the initial condition is given as $X(t_0) \sim N(\mu_0, \Sigma_0)$, Eq. (2) becomes

$$X(t) \sim N\left(\mu(t), \Sigma(t)\right)$$
$$\mu(t) = \Phi(t)\left(\Phi(t_0)^{-1}\mu_0 + \int_{t_0}^{t} \Phi(\tau)^{-1}\tilde{u}(\tau)d\tau\right), \ \Sigma(t) = \Phi(t)\Sigma'(t)\Phi(t)^\top \tag{4}$$

where $\Sigma'(t) = \Phi(t_0)^{-1}\Sigma_0(\Phi(t_0)^{-1})^\top + \int_{t_0}^{t} \Phi(\tau)^{-1}Q(\Phi(\tau)^{-1})^\top d\tau$ .

Note that if $\forall i : \lambda_i < 0$ and $\tilde{u} \equiv 0$, we have $\mu(t) \xrightarrow{t \to \infty} 0$ (stable system). In addition, if $\lambda$ is complex, Eq. (4) may produce a complex solution; Appendix C explains how to use a careful parameterization to only calculate the real solutions.

## 3 Problem Setup: Sparsely-Observable SDE

We focus on online sequential prediction of a process $Y(t) \in \mathbb{R}^m$. To predict $Y(t_0)$ at a certain $t_0$, we can use noisy observations $\hat{Y}(t)$ (at given times $t < t_0$), as well as a control signal $u(t) \in \mathbb{R}^k$ ($\forall t < t_0$); offline data of $Y$ and $u$ from other sequences; and one sample of contextual information $C \in \mathbb{R}^{d_c}$ per sequence (capturing properties of the whole sequence). **The dynamics of $Y$ are unknown and may vary between sequences**. For example, sequences may represent different patients, each with its own dynamics; $C$ may represent patient information; and the objective is "zero-shot" learning upon arrival of a sequence of any new patient. In addition, **the observations within a sequence are**

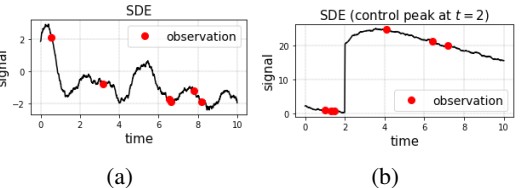

(a)          (b)

Figure 1: Samples of sparsely observed SDEs: the Brownian noise and the sparse observations pose a major challenge for learning the underlying SDE dynamics. Efficient learning from external trajectories data is required, as the current trajectory often does not contain sufficient observations.

**both irregular and sparse**: they are received at arbitrary points of time, and are sparse in comparison to the required prediction frequency (i.e., continuous forecasting, as illustrated in Fig. 1).

To model the problem, we assume the observations $Y(t)$ to originate from an unobservable latent process $X(t) \in \mathbb{R}^n$ (where $n > m$ is a hyperparameter). More specifically:

$$dX(t) = F_C\big(X(t), u(t)\big), \ Y(t) = X(t)_{1:m}, \ \hat{Y}(t) = Y(t) + \nu_C(t) \tag{5}$$

where $F_C$ is a stochastic dynamics operator (which may depend on the context $C$); $Y$ is simply the first $m$ coordinates of $X$; $\hat{Y}$ is the corresponding observation; and $\nu_C(t)$ is its i.i.d Gaussian noise with zero-mean and (unknown) covariance $R_C \in \mathbb{R}^{m \times m}$ (which may also depend on $C$). Our goal is to predict $Y$, where the dynamics $F_C$ are unknown and the latent subspace of $X$ is unobservable. In cases where data of $Y$ is not available, we measure our prediction accuracy against $\hat{Y}$. The control $u(t)$ is modeled separately from $\hat{Y}$, is not part of the prediction objective, and does not depend on $X$.

## 4 Neural Eigen-SDE Algorithm

**Model:** In this section, we introduce the Neural Eigen-SDE algorithm (NESDE, shown in Algorithm 1 and Fig. 2). NESDE predicts the signal $Y(t)$ of Eq. (5) continuously at any required point of time $t$. It relies on a piecewise linear approximation which reduces Eq. (5) into Eq. (1):

$$\forall t \in \mathcal{I}_i : \ dX(t) = [A_i \cdot (X(t) - \alpha) + B \cdot u(t)] + dW(t) \tag{6}$$

where $\mathcal{I}_i = (t_i, t_{i+1})$ is a time interval, $dW$ is a Brownian noise with covariance matrix $Q_i$, and $A_i \in \mathbb{R}^{n \times n}, B \in \mathbb{R}^{n \times k}, Q_i \in \mathbb{R}^{n \times n}, \alpha \in \mathbb{R}^n$ form the linear dynamics model corresponding to the

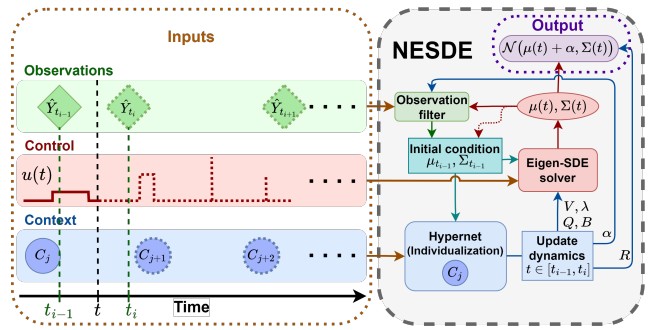
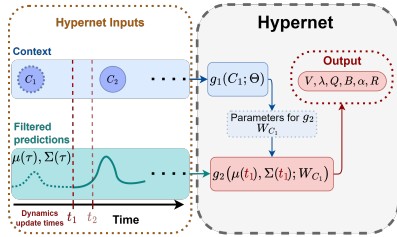

Figure 2: NESDE algorithm: Hypernet determines the SDE parameters; Eigen-SDE solver uses them to predict the next time-interval; the filter updates the state upon arrival of a new observation, which initiates a new interval. For more frequent updates of the dynamics, the initial condition becomes the last prediction (dotted arrow).

Figure 3: Zoom into the individualization module of NESDE: Hypernet uses the context and the estimated state to determine the SDE parameters for the solver.

interval $\mathcal{I}_i$. In terms of Eq. (1), we substitute $A := A_i$ and $\tilde{u} := Bu - A_i\alpha$. Note that if $A_i$ is a stable system and $u \equiv 0$, the asymptotic state is $\mu(t) \xrightarrow{t \to \infty} \alpha$. To solve Eq. (6) within every $\mathcal{I}_i$, NESDE has to learn the parameters $\{A_i, Q_i\}_i, \alpha, B$.

The end of $\mathcal{I}_{i-1}$ typically represents one of two events: either an update of the dynamics $A$ (allowing the piecewise linear dynamics), or the arrival of a new observation. A new observation at time $t_i$ triggers an update of $X(t_i)$ according to the conditional distribution $X(t_i)|\hat{Y}(t_i)$ (this is a particular case of Kalman filtering, as shown in Appendix A). Then, the prediction continues for $\mathcal{I}_i$ according to Eq. (6). Note that once $X(t_0)$ is initialized to have a Normal distribution, it remains Normally-distributed throughout both the process dynamics (Eq. (4)) and the observations filtering (Appendix A). This allows NESDE to efficiently capture the distribution of $X(t)$, where the estimated covariance represents the uncertainty.

**Eigen-SDE solver (ESDE) – spectral dynamics representation:** A key feature of NESDE is that $A_i$ is only represented implicitly through the parameters $V, \lambda$ defining its eigenfunction $\Phi(t)$ of Eq. (3) (we drop the interval index $i$ with a slight abuse of notation). The spectral representation allows Eq. (4) to solve $X(t)$ analytically for any $t \in \mathcal{I}_i$ at once: the predictions are not limited to predefined times, and do not require recursive iterations with constant time-steps. This is particularly useful in the sparsely-observable setup of Section 3, as it lets numerous predictions be made *at once* without being "interrupted" by a new measurement.

The calculation of Eq. (4) requires efficient integration. Many SDE solvers apply recursive numeric integration [Chen et al., 2018, De Brouwer et al., 2019]. In NESDE, however, thanks to the spectral decomposition, the integration only depends on known functions of $t$ instead of $X(t)$ (Eq. (4)), hence recursion is not needed, and the computation can be paralleled. Furthermore, if the control $u$ is constant over an interval $\mathcal{I}_i$ (or has any other analytically-integrable form), Appendix B shows how to calculate the integration *analytically*. Piecewise constant $u$ is common, for example, when the control is updated along with the observations.

In addition to simplifying the calculation, the spectrum of $A_i$ carries significant meaning about the dynamics. For example, negative eigenvalues correspond to a stable solution, whereas imaginary ones indicate periodicity, as demonstrated in Fig. 4. Note that the (possibly-complex) eigenvalues must be constrained to represent a real matrix $A_i$. The constraints and the calculations in the complex space are detailed in Appendix C.

Note that if the process $X(t)$ actually follows the piecewise linear model of Eq. (6), and the model parameters are known correctly, then the Eigen-SDE solver trivially returns the optimal predictions. The complete proposition and proof are provided in Appendix D.

**Proposition 1** (Eigen-SDE solver optimality). *If $X(t)$ follows Eq. (6) with the same parameters used by the Eigen-SDE solver, then under certain conditions, the solver prediction at any point of time optimizes both the expected error and the expected log-likelihood.*

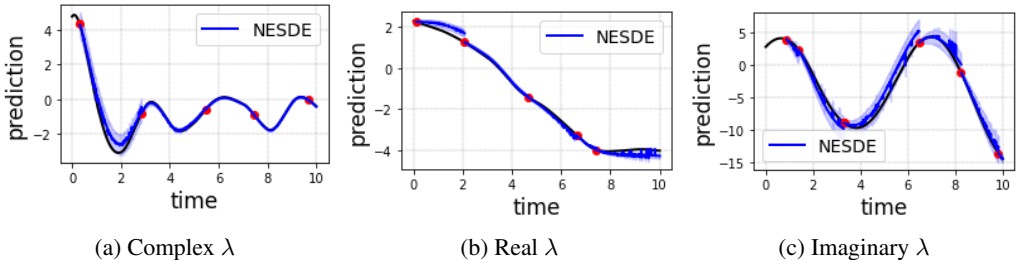

(a) Complex $\lambda$          (b) Real $\lambda$          (c) Imaginary $\lambda$

Figure 4: Sample trajectories with different types of dynamics (and random control signal that is not displayed). As demonstrated in Appendix E.5, NESDE directly estimates the dynamics $\lambda$ and reveals their type.

*Proof Sketch.* If the process $X(t)$ follows Eq. (6), the Eigen-SDE solver output corresponds to the true distribution $X(t) \sim N(\mu(t), \Sigma(t))$ for any $t$. Thus, both the expected error and the expected log-likelihood are optimal.     □

**Updating solver and filter parameters:** NESDE provides the parameters $V, \lambda, Q, B, \alpha$ to the Eigen-SDE solver, as well as the noise $R$ to the observation filter. As NESDE assumes a *piecewise* linear model, it separates the time into intervals $\mathcal{I}_i = (t_i, t_{i+1})$ (the interval length is a hyperparameter), and uses a dedicated model to predict new parameters at the beginning $t_i$ of every interval.

The model receives the current state $X(t_i)$ and the contextual information $C$, and returns the parameters for $\mathcal{I}_i$. Specifically, we use Hypernet [Ha et al., 2016]: one neural network $g_1$ with network parameters $\Theta$, returns the weights

---

**Algorithm 1** NESDE

**Input:** context $C$; control signal $u(t)$; update times $\mathcal{I} \in \mathbb{R}^T$; prediction times $\{P_{\mathcal{I}_i}\}_{\mathcal{I}_i \in \mathcal{I}}$
**Initialize:** $\mu, \Sigma, \alpha, R \leftarrow \mathbf{Prior}(C)$
**for** $\mathcal{I}_i$ in $\mathcal{I}$: **do**
    $V, \lambda, Q, B, \alpha, R \leftarrow \mathbf{Hypernet}(C, \mu, \Sigma)$
    **for** $t$ in $P_{\mathcal{I}_i}$ **do**
        $\mu_t, \Sigma_t \leftarrow \mathbf{ESDE}(\mu, \Sigma, u, t; V, \lambda, Q, B)$   *// Eq. (4)*
        **predict:** $\tilde{Y}_t \sim \mathcal{N}(\mu_t + \alpha, \Sigma_t + R)$
        **if** given observation $\hat{Y}_t$ **then**
            $\mu, \Sigma \leftarrow \mathbf{Filter}(\mu_t, \Sigma_t, R, \hat{Y}_t)$   *// Appendix A*
        **end if**
    **end for**
**end for**

---

of a second network $g_2$, i.e., $W = g_1(C; \Theta)$. Then, $g_2$ maps the current state estimation to a set of dynamics parameters: $(V, \lambda, Q, B, \alpha, R) = g_2(X; W) = g_2(X; g_1(C; \Theta))$ (Fig. 3). The dynamics $g_2(X; W)$ may be updated more often than the individualized weights $W = g_1(C; \Theta)$, as the context $C$ may change with a lower frequency. For the initial state, where $X$ is unavailable, we learn a *state prior* from $C$ by a dedicated network; this prior helps NESDE to function as a "zero-shot" model.

The Hypernet module implementation gives us control over the non-linearity and non-stationarity of the model. The importance of the Hypernet module for individualized modeling is demonstrated in an ablation test in Appendix E.1. In our current implementation, only $V, \lambda, Q$ are renewed every time interval. $\alpha$ (asymptotic signal) and $R$ (observation noise) are only predicted once per sequence, as we assume they are independent of the state. The control mapping $B$ is assumed to be a global parameter.

**Training:** The learnable parameters of NESDE are the control mapping $B$ and Hypernet's parameters $\Theta$ (which in turn determine the rest of the parameters). To optimize them, the training relies on a dataset of sequences of control signals $\{u_{seq}(t_j)\}_{seq,j}$ and (sparser and possibly irregular) states and observations $\{(Y_{seq}(t_j), \hat{Y}_{seq}(t_j))\}_{seq,j}$ (if $Y$ is not available, we use $\hat{Y}$ instead as the training target).

The latent space dimension $n$ and the model-update frequency $\Delta t$ are determined as hyperparameters. Then, we use the standard Adam optimizer [Diederik P. Kingma, 2015] to optimize the parameters with respect to the loss $NLL(j) = -\log P(Y(t_j)|\mu(t_j), \Sigma(t_j))$ (where $\mu, \Sigma$ are predicted by NESDE sequentially from $u, \hat{Y}$). Each training iteration corresponds to a batch of sequences of data, where the $NLL$ is aggregated over all the samples of the sequences. Note that our supervision for the training is limited to the times of the observations, even if we wish to make more frequent predictions in inference.

Table 2: Test errors in the irregular synthetic benchmarks, estimated over 5 seeds and 1000 test trajectories per seed, with standard deviation calculated across seeds.

| Model | Complex dynamics eigenvalues | | Real dynamics eigenvalues | |
|---|---|---|---|---|
| | MSE | OOD MSE | MSE | OOD MSE |
| LSTM | $0.23 \pm 0.001$ | $0.589 \pm 0.02$ | $0.381 \pm 0.002$ | $2.354 \pm 0.84$ |
| GRU-ODE-Bayes | $0.182 \pm 0.0004$ | $0.361 \pm 0.044$ | $\mathbf{0.219 \pm 0.0004}$ | $0.355 \pm 0.005$ |
| CRU | $0.233 \pm 0.0054$ | $0.584 \pm 0.009$ | $0.231 \pm 0.001$ | $0.541 \pm 0.026$ |
| NESDE (ours) | $\mathbf{0.176 \pm 0.0001}$ | $\mathbf{0.178 \pm 0.001}$ | $0.222 \pm 0.0005$ | $\mathbf{0.332 \pm 0.005}$ |

As demonstrated below, the unique architecture of NESDE provides effective regularization and data efficiency (due to piecewise linearity), along with tunable expressiveness (neural updates with controlled frequency). Yet, it is important to note that the piecewise linear SDE operator does limit the expressiveness of the model (e.g., in comparison to other neural-ODE models). Further, NESDE is only optimal under a restrictive set of assumptions, as specified in Proposition 1.

## 5 Synthetic Data Experiments

In this section, we test three main aspects of NESDE: (1) prediction from partial and irregular observations, (2) robustness to out-of-distribution control (OOD), and (3) sample efficiency. We experiment with data of a simulated stochastic process, designed to mimic partially observable medical processes with indirect control.

The simulated data includes trajectories of a 1-dimensional signal $Y$, with noiseless measurements at random irregular times. The goal is to predict the future values of $Y$ given its past observations. However, $Y$ is mixed with a latent (unobservable) variable, and they follow linear dynamics with both decay and periodicity (i.e., complex dynamics eigenvalues). In addition, we observe a control signal that affects the latent variable (hence affects $Y$, but only indirectly through the dynamics). The control negatively correlated with the observations: $u_t = b_t - 0.5 \cdot Y_t$. $b_t \sim U[0, 0.5]$ is a piecewise constant additive noise (changing 10 times per trajectory).

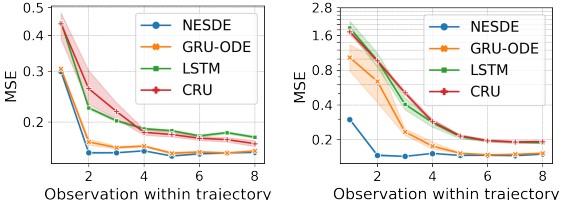

(a) Same control distribution (b) Out of distribution control

Figure 5: MSE vs. number of observations so far in the trajectory, in the complex dynamics setting, for: (a) standard test set, and (b) test set with out-of-distribution control policy. 95% confidence intervals are calculated over 5 seeds.

As baselines for comparison, we choose recent SDE-based methods that provide Bayesian uncertainty estimation: **GRU-ODE**-Bayes [De Brouwer et al., 2019] and **CRU** [Schirmer et al., 2022] (despite terminology, both implement an SDE rather than ODE). In these methods, concatenating the control signal to the observation results in poor learning, as the control becomes part of the model output and dominates the loss function. To enable effective learning for the baselines, we mask-out the control from the loss. Additionally, we design a dedicated **LSTM** model that supports irregular predictions, as described in Appendix F.2.

**Out-of-distribution control (OOD):** We simulate two benchmarks – one with *complex* eigenvalues and another with *real* eigenvalues (no periodicity). We train all models on a dataset of 1000 random trajectories, and test on a separate dataset – with different trajectories that follow the *same distribution*. In addition, we use an *OOD* test dataset, where the control is positively correlated with the observations: $u_t = b_t + 0.5 \cdot Y_t$. This can simulate, for example, forecasting of the same biochemical process after changing the medicine dosage policy.

Table 2 and Fig. 5a summarize the prediction errors. Before changing the control policy, NESDE achieves the best accuracy in the complex dynamics, and is on par with GRU-ODE-Bayes in the real dynamics. Notice that CRU, which relies on a real-valued linear model in latent space, is indeed sub-optimal under the complex dynamics, compared to NESDE and GRU-ODE-Bayes. The LSTM presents high errors in both benchmarks.

Once the control changes, all models naturally deteriorate. Yet, NESDE presents the smallest deterioration and best accuracy in the OOD test datasets – for both complex and real dynamics. In particular, NESDE provides a high prediction accuracy after mere 2 observations (Fig. 5b), making it a useful zero-shot model. The robustness to the modified control policy can be attributed to the model of NESDE in Eq. (6), which decouples the control from the observations.

In a similar setting in Appendix E.6, the control $u$ used in the training data has continuous knowledge of $Y$. Since the model only observes $Y$ in a limited frequency, $u$ carries additional information about $Y$. This results in extreme overfitting and poor generalization to different control policies – for all methods except for NESDE, which maintains robust OOD predictions in this challenging setting.

**Sample efficiency:** We train each method over datasets with different number of trajectories. Each model is trained on each dataset separately until convergence. As shown in Fig. 6, NESDE achieves the best test accuracy for every training dataset, and learns reliably even from as few as 100 trajectories. The other methods deteriorate significantly in the smaller datasets. Note that in the real dynamics, LSTM fails regardless of the amount of data, as also reflected in Table 2.

GRU-ODE-Bayes achieves the best sample efficiency among the baselines. In Appendix E.2, we use **a benchmark from the study of GRU-ODE-Bayes itself** [De Brouwer et al., 2019], and demonstrate the superior sample efficiency of NESDE in that benchmark as well. Appendix E.3 extends the notion of sample efficiency to sparse trajectories: for a constant number of training trajectories, it reduces the number of observations per trajectory. NESDE demonstrates high robustness to the amount of data in that setting as well.

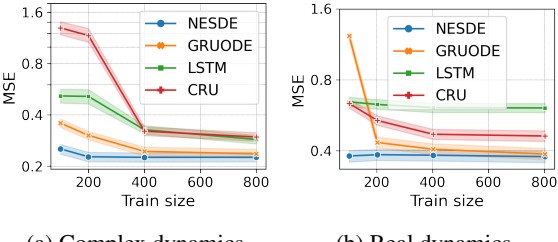

(a) Complex dynamics      (b) Real dynamics

Figure 6: Test MSE vs. train data size. 95% confidence intervals are calculated over 1000 test trajectories.

**Regular LSTM:** Appendix E.4 extends the experiments for regular data with constant time-steps. In the regular setting, LSTM provides competitive accuracy when observations are dense. However, if the signal is not observed every time-step, LSTM deteriorates significantly in unobserved time-steps (Fig. 12c), possibly because gradients have to be propagated over many steps. Hence, even in regular settings, LSTM struggles to provide predictions more frequent than the measurements.

# 6 Medication Dosing Regimes

As discussed in Section 1, many medical applications could potentially benefit from ODE-based methods. Specifically, we address medication dosing problems, where observations are often sparse, the dosing is a control signal, and uncertainty estimation is crucial. We test **NESDE** on two such domains, against the same baselines as in Section 5 (**GRU-ODE-Bayes**, **CRU** and an irregular **LSTM**). We also add a **naive** model with "no-dynamics" (predicts the last observed value).

The benchmarks in this section were derived from the MIMIC-IV dataset [Johnson et al., 2020]. Typically to electronic health records, the dataset contains a vast amount of side-information (e.g., weight and heart rate). We use some of this information as an additional input – for each model according to its structure (context-features for the hyper-network of NESDE, covariates for GRU-ODE-Bayes, state variables for CRU, and embedding units for the LSTM). Some context features correspond to online measurements and are occasionally updated. In both domains, we constraint the process eigenvalues $\lambda$ to be negative, to reflect the stability of the biophysical processes. Indeed, the spectral representation of NESDE provides us with a natural way to incorporate such domain knowledge, which often cannot be used otherwise. For all models, in both domains, we use a 60-10-30 train-validation-test data partition. See more implementation details in Appendix F.

## 6.1 Unfractionated Heparin Dosing

Unfractionated Heparin (UH) is a widely used anticoagulant drug. It may be given in a continuous infusion to patients with life-threatening clots and works by interfering with the normal coagulation cascade. As the effect is not easily predicted, the drug's actual activity on coagulation is traditionally

Table 3: Test mean square errors (MSE) and negative log-likelihood (NLL, for models that provide probabilistic prediction) in the medication-dosing benchmarks.

| Model | UH Dosing | | Vancomycin Dosing | |
|---|---|---|---|---|
| | MSE | NLL | MSE | NLL |
| **Naive** | $613.3 \pm 13.48$ | – | $112.2 \pm 16.4$ | – |
| **LSTM** | $482.1 \pm 6.52$ | – | $92.89 \pm 11.3$ | – |
| **GRU-ODE-Bayes** | $491 \pm 6.88$ | $4.52 \pm 0.008$ | $80.54 \pm 11.8$ | $6.38 \pm 0.12$ |
| **CRU** | $450.4 \pm 8.27$ | $4.49 \pm 0.012$ | $76.4 \pm 12.8$ | $3.87 \pm 0.2$ |
| **NESDE (ours)** | $\mathbf{411.2 \pm 7.39}$ | $\mathbf{4.43 \pm 0.01}$ | $\mathbf{70.71 \pm 12.3}$ | $\mathbf{3.69 \pm 0.13}$ |

monitored using a lab test performed on a blood sample from the patient: activated Partial Thromboplastin Time (aPTT) test. The clinical objective is to keep the aPTT value in a certain range. The problem poses several challenges: different patients are affected differently; the aPTT test results are delayed; monitoring and control are required in higher frequency than measurements; and deviations of the aPTT from the objective range may be fatal. In particular, underdosed UH may cause clot formation and overdosed UH may cause an internal bleeding [Landefeld et al., 1987]. Dosage rates are manually decided by a physician, using simple protocols and trial-and-error over significant amounts of time. Here we focus on continuous prediction as a key component for aPTT control.

Following the preprocessing described in Appendix F.1, the MIMIC-IV dataset derives 5866 trajectories of a continuous UH control signal, an irregularly-observed aPTT signal (whose prediction is the goal), and 42 context features. It is known that UH does not affect the coagulation time (aPTT) directly (but only through other unobserved processes, Delavenne et al. [2017]); thus, we mask the control mapping $B$ to have no direct effect on the aPTT metric, but only on the latent variable (which can be interpreted as the body UH level). The control (UH) and observations (aPTT) are one-dimensional ($m = 1$), and we set the whole state dimension to $n = 4$.

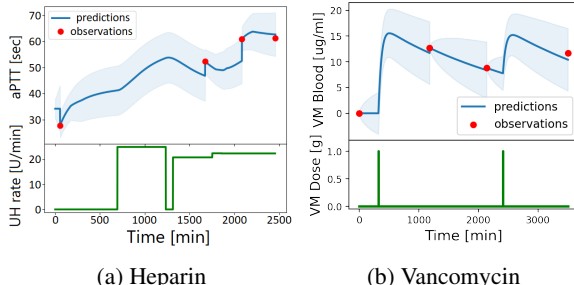

(a) Heparin  (b) Vancomycin

Figure 7: A sample of patients from (a) the UH dosing dataset, and (b) the VM dosing dataset. The lower plots correspond to medication dosage (UH in (a) and VM in (b)). The upper plots correspond to the continuous prediction of NESDE (aPTT levels in (a) and VM concentration in (b)), with 95% confidence intervals. In both settings, the prediction at every point relies on all the observations up to that point.

## 6.2 Vancomycin Dosing

Vancomycin (VM) is an antibiotic that has been in use for several decades. However, the methodology of dosing VM remains a subject of debate in the medical community [Rybak et al., 2009], and there is a significant degree of variability in VM dynamics among patients [Marsot et al., 2012]. The dosage of VM is critical; it could become toxic if overdosed [Filippone et al., 2017], and ineffective as an antibiotic if underdosed. The concentration of VM in the blood can be measured through lab test, but these tests are often infrequent and irregular, which fits into our problem setting.

Here, the goal is to predict the VM concentration in the blood at any given time, where the dosage and other patient measurements are known. Following the preprocessing described in Appendix F.1, the dataset derives 3564 trajectories of VM dosages at discrete times, blood concentration of VM ($m = 1$) at irregular times, and similarly to Section 6.1, 42 context features. This problem is less noisy than the UH dosing problem, as the task is to learn the direct dynamics of the VM concentration, and not the effects of the antibiotics. The whole state dimension is set to $n = 2$, and we also mask the control mapping $B$ to have no direct effect on the VM concentration, where the latent variable that directly affected could be viewed as the amount of drug within the *whole body*, which in turn affects the actual VM concentration in the *blood*.

## 6.3 Results

Fig. 7 displays sample trajectories predicted by NESDE in both domains. As summarized in Table 3, NESDE outperforms the other baselines in both UH and VM dosing tasks, in terms of both square errors (MSE) and likelihood (NLL). For the UH dosing problem, Fig. 8 also presents the errors

vs. prediction horizon (the time passed since the last observation). Evidently, **NESDE provides the best accuracy in all the horizons**. Its smoothness allows NESDE to obtain high accuracy in short horizons, and its robustness allows **generalization to out-of-distribution horizons**: while most of the data corresponds to horizons of 5-7 hours (see Fig. 14 in the appendix), NESDE provides reliable prediction at other horizons as well. By contrast, LSTM and GRU-ODE-Bayes have difficulty with short horizons; they only become competitive with the *naive* model after 6 hours. CRU provides more robust predictions, but is still outperformed by NESDE.

Despite the large range of aPTT levels in the data (e.g., the top 5% are above $100s$ and the bottom 5% are below $25s$), 50% of all the predictions have errors lower than $12.4s$ – an accuracy level that is considered clinically safe. Fig. 8 shows that indeed, up to 3 hours after the last lab test, the average error is smaller than $10s$.

**Running time**: All Heparin and Vancomycin experiments were run on a single Ubuntu machine with eight i9-10900X CPU cores and Nvidia's RTX A5000 GPU. A single training process of NESDE required 8-24 hours (with variance mostly attributed to early stopping criterion). LSTM required similar running times (6-24 hours), whereas GRU-ODE-Bayes and CRU required 36-72 hours to train.

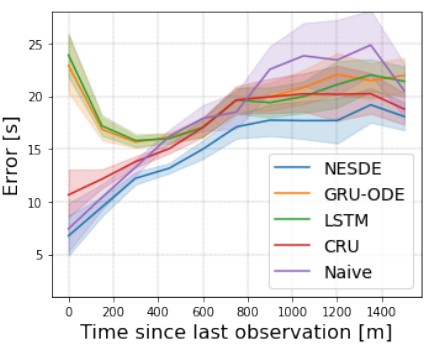

Figure 8: aPTT prediction errors in the UH problem, vs. the time passed since the last aPTT test.

## 7   Conclusion

Motivated by medical forecasting and control problems, we characterized a set of challenges in modeling continuous dosing dynamics: sample efficiency, uncertainty estimation, personalized modeling, continuous inference and generalization to different control policies. To address these challenges, we introduced the novel NESDE algorithm, based on a stochastic differential equation with spectral representation. We demonstrated the reliability of NESDE in a variety of synthetic and real data experiments, including high noise, little training data, contextual side-information and out-of-distribution control signals. In addition, NESDE demonstrated high prediction accuracy after as few as 2 observations, making it a useful zero-shot model.

We applied NESDE to two real-life high-noise medical problems with sparse and irregular measurements: (1) blood coagulation forecasting in Heparin-treated patients, and (2) Vancomycin levels prediction in patients treated by antibiotics. In both problems, NESDE significantly improved the prediction accuracy compared to alternative methods. This demonstrates NESDE's advantage in continuous forecasting of controlled SDE systems, which include medication dosing and controlled chemical and biological processes.

As demonstrated in the experiments, NESDE provides robust, reliable and uncertainty-aware continuous *forecasting*. This paves the way to development of *decision making* in continuous high-noise decision processes, including medical treatment, finance and operations management. Future research may address medical optimization via both control policies (e.g., to control medication dosing) and sampling policies (to control measurements timing, e.g., of blood tests).

**Acknowledgements:** This research was supported by VATAT Fund to the Technion Artificial Intelligence Hub (Tech.AI).

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

# Appendices

## Table of Contents

## A Observation Filtering: The Conditional Distribution and the Relation to Kalman Filtering

As described in Section 4, the NESDE algorithm keeps an estimated Normal distribution of the system state $X(t)$ at any point of time. The distribution develops continuously through time according to the dynamics specified by Eq. (6), except for the discrete times where an observation $\hat{Y}(t)$ is received: in every such point of time, the $X(t)$ estimate is updated to be the conditional distribution $X(t)|\hat{Y}(t)$.

**Calculating the conditional Normal distribution:** The conditional distribution can be derived as follows. Recall that $X \sim N(\mu, \Sigma)$ (we remove the time index $t$ as we focus now on filtering at a single point of time). Denote $X = (Y, Z)^\top$ where $Y \in \mathbb{R}^m$ (similarly to Eq. (5)) and $Z \in \mathbb{R}^{n-m}$; and similarly, $\mu = (\mu_Y, \mu_Z)^\top$ and

$$\Sigma = \begin{pmatrix} \Sigma_{YY} & \Sigma_{YZ} \\ \Sigma_{ZY} & \Sigma_{ZZ} \end{pmatrix}$$

First consider a noiseless observation ($R = 0$): then according to Eaton [1983], the conditional distribution $X|Y = \hat{Y}$ is given by $X = (Y, Z)^\top$, $Y = \hat{Y}$ and $Z \sim N(\mu'_Z, \Sigma'_{ZZ})$, where

$$\mu'_Z := \mu_Z + \Sigma_{ZY}\Sigma_{YY}^{-1}(\hat{Y} - \mu_Y)$$

$$\Sigma'_{ZZ} := \Sigma_{ZZ} - \Sigma_{ZY}\Sigma_{YY}^{-1}\Sigma_{YZ}$$

In the general case of $R \neq 0$, we can redefine the state to include the observation explicitly: $\tilde{X} = (\hat{Y}, X)^\top = (\hat{Y}, Y, Z)^\top$, where $\tilde{\mu}, \tilde{\Sigma}$ of $\tilde{X}$ are adjusted by $\mu_{\hat{Y}} = \mu_y$, $\Sigma_{\hat{Y}\hat{Y}} = \Sigma_{YY} + R$, $\Sigma_{\hat{Y}Y} = R$ and $\Sigma_{\hat{Y}Z} = \Sigma_{YZ}$. Then, the conditional distribution can be derived as in the noiseless case above, by simply considering the new observation as a noiseless observation of $\tilde{X}_{1:m} = \hat{Y}$.

**The relation to the Kalman filtering:** The derivation of the conditional distribution is equivalent to the filtering step of the Kalman filter [Kalman, 1960], where the (discrete) model is

$$X_{t+1} = A \cdot X_t + \omega_t \qquad (\omega_t \sim N(0, Q))$$

$$\hat{Y}_t = H \cdot X_t + \nu_t \qquad (\nu_t \sim N(0, R)),$$

Our setup can be recovered by substituting the following observation model $H \in \mathbb{R}^{m \times n}$, which observes the first $m$ coordinates of $X$ and ignores the rest:

$$H = \begin{pmatrix} 1 & & & & 0 & \dots & 0 \\ & 1 & & & & & \\ & & \dots & & | & | & | \\ & & & 1 & & & \\ & & & 1 & 0 & \dots & 0 \end{pmatrix}$$

and the Kalman filtering step is then

$$K := \Sigma H^\top (H\Sigma H^\top + R)^{-1}$$

$$\mu' := \mu + K(\hat{Y} - H\mu)$$

$$\Sigma' := \Sigma - KH\Sigma$$

Note that while the standard Kalman filter framework indeed supports the filtering of distributions upon arrival of a new observation, its progress through time is limited to discrete and constant time-steps (see the model above), whereas our SDE-based model can directly make predictions to any arbitrary future time $t$.

## B Integrator Implementation

Below, we describe the implementation of the integrator of the Eigen-SDE solver mentioned in Section 4.

**Numerical integration given $u(t)$:** In the presence of an arbitrary (continuous) control signal $u(t)$, it is impossible to compute the integral that corresponds with $u(t)$ (Eq. (2)) analytically. On the

other hand, $u(t)$ is given in advance, and the eigenfunction, $\Phi(t)$, is a known function that can be calculated efficiently at any given time. By discretizing the time to any fixed $\Delta t$, one could simply replace the integral by a sum term

$$\int_{t_0}^{t} \Phi(\tau)^{-1}u(\tau)d\tau \approx \sum_{i=0}^{\frac{t-t_0}{\Delta t}} \Phi(t_0 + i \cdot \Delta t)u(t_0 + i \cdot \Delta t)\Delta t$$

while this sum represent $\frac{t-t_0}{\Delta t}$ calculations, it can be computed efficiently, as it does not require any recursive computation, as both $\Phi(t)$ and $u(t)$ are pre-determined, known functions. Each element of the sum is independent of the other elements, and thus the computation could be parallelized.

**Analytic integration:** The control $u$ is often constant over any single time-interval $\mathcal{I}$ (e.g., when the control is piecewise constant). In such cases, for a given interval $\mathcal{I} = [t_0, t]$ in which $u(t) = u_{\mathcal{I}}$, the integral could be solved analytically:

$$\int_{t_0}^{t} \Phi(\tau)^{-1}u(\tau)d\tau = \int_{t_0}^{t} e^{-\Lambda\tau}V^{-1}u_{\mathcal{I}}d\tau = \int_{t_0}^{t} e^{-\Lambda\tau}d\tau V^{-1}u_{\mathcal{I}} = \frac{1}{\Lambda}\left(e^{-\Lambda t_0} - e^{-\Lambda t}\right)V^{-1}u_{\mathcal{I}}$$

one might notice that for large time intervals this form is numerically unstable, to address this issue, note that this integral is multiplied (Eq. (2)) by $\Phi(t) = Ve^{\Lambda t}$, hence we stabilize the solution with the latter exponent:

$$\Phi(t)\frac{1}{\Lambda}\left(e^{-\Lambda t_0} - e^{-\Lambda t}\right)V^{-1}u_{\mathcal{I}} = V\frac{1}{\Lambda}\left(e^{\Lambda(t-t_0)} - e^{\Lambda(t-t)}\right)V^{-1}u_{\mathcal{I}} = V\frac{1}{\Lambda}\left(e^{\Lambda(t-t_0)} - 1\right)V^{-1}u_{\mathcal{I}}$$

to achieve a numerically stable computation.

In addition to the integral over $u(t)$, we also need to calculate the integral over $Q$ (Eq. (4)). In this case, $Q$ is constant, and the following holds;

$$\int_{t_0}^{t} \Phi(\tau)^{-1}Q(\Phi(\tau)^{-1})^{\top}d\tau = \int_{t_0}^{t} e^{-\Lambda\tau}V^{-1}Q(V^{-1})^{\top}(e^{-\Lambda\tau})^{\top}d\tau = V^{-1}Q(V^{-1})^{\top} \circ \int_{t_0}^{t} e^{-\tilde{\Lambda}\tau}d\tau$$

where $\circ$ denotes the Hadamard product, and

$$\tilde{\Lambda} = \begin{pmatrix} 2\lambda_1 & \cdots & \lambda_1 + \lambda_n \\ \vdots & \ddots & \\ \lambda_n + \lambda_1 & \cdots & 2\lambda_n \end{pmatrix}$$

In this form, it is possible to solve the integral analytically, similarly to the integral of the control signal, and again, we use the exponent term from $\Phi(t)$ to obtain a numerically stable computation.

## C  The Dynamics Spectrum and Complex Eigenfunction Implementation

The form of the eigenfunction matrix as presented in Section 2 is valid for real eigenvalues. Complex eigenvalues induce a slightly different form; firstly, they come in pairs, i.e., if $z = a + bi$ is an eigenvalue of $A$ (Eq. (1)), then $\bar{z} = a - bi$ (the complex conjugate of $z$) is an eigenvalue of $A$. The corresponding eigenvector of $z$ is complex as well, denote it by $v = v_{real} + v_{im}i$, then $\bar{v}$ (the complex conjugate of $v$) is the eigenvector that correspond to $\bar{z}$. Secondly, the eigenfunction matrix takes the form:

$$\Phi(t) = e^{at}\left(v_{real} \cdot cos(bt) - v_{im} \cdot sin(bt) \quad v_{im} \cdot cos(bt) + v_{real} \cdot sin(bt)\right)$$

For brevity, we consider only the elements that correspond with $z, \bar{z}$. To parametrize this form, we use the same number of parameters (each complex number need two parameters to represent, but since they come in pairs with their conjugates we get the same overall number) which are organized differently. Mixed eigenvalues (e.g., both real and complex) induce a mixed eigenfunction that is a concatenation of the two forms. Since the complex case requires a different computation, we leave the number of complex eigenvalues to be a hyperparameter. Same as for the *real* eigenvalues setting, it is possible to derive an analytical computation for the integrals. Here, it takes a different form, as

the complex eigenvalues introduce trigonometric functions to the eigenfunction matrix. To describe the analytical computation, first notice that:

$$\Phi(t) = e^{at} \begin{pmatrix} | & | \\ v_{real} & v_{im} \\ | & | \end{pmatrix} \begin{pmatrix} cos(bt) & sin(bt) \\ -sin(bt) & cos(bt) \end{pmatrix}$$

and thus:

$$\Phi(t)^{-1} = e^{-at} \begin{pmatrix} cos(bt) & -sin(bt) \\ sin(bt) & cos(bt) \end{pmatrix} \begin{pmatrix} | & | \\ v_{real} & v_{im} \\ | & | \end{pmatrix}^{-1}$$

Note that here we consider a two-dimensional SDE, for the general case the trigonometric matrix is a block-diagonal matrix, and the exponent becomes a diagonal matrix in which each element repeats twice. It is clear that similarly to the real eigenvalues case, the integral term that includes $u$ (as shown above) can be decomposed, and it is possible to derive an analytical solution for an exponent multiplied by sine or cosine. One major difference is that here we use matrix product instead of Hadamard product. The integral over $Q$ becomes more tricky, but it can be separated and computed as well, with the assistance of basic linear algebra (both are implemented in our code).

## D  Solver Analysis

Below is a more complete version of Proposition 1 and its proof.

**Proposition 2** (Eigen-SDE solver optimality: complete formulation). Let $X(t)$ be a signal that follows Eq. (6) for any time interval $\mathcal{I}_i = [t_i, t_{i+1}]$, and $u(t)$ a control signal that is constant over $\mathcal{I}_i$ for any $i$. For any $i$, consider the Eigen-SDE solver with the parameters corresponding to Eq. (6) (for the same $\mathcal{I}_i$). Assume that the first solver ($i = 0$) is initialized with the true initial distribution $X(0) \sim N(\mu_0, \Sigma_0)$, and for $i \geq 1$, the $i$'th solver is initialized with the $i-1$'th output, along with an observation filter if an observation was received. For any interval $i$ and any time $t \in \mathcal{I}_i$, consider the prediction $\tilde{X}(t) \sim N(\mu(t), \Sigma(t))$ of the solver. Then, $\mu(t)$ minimizes the expected square error of the signal $X(t)$, and $\tilde{X}(t)$ maximizes the expected log-likelihood of $X(t)$.

*Proof.* We prove by induction over $i$ that for any $i$ and any $t \in \mathcal{I}_i$, $\tilde{X}(t)$ corresponds to the true distribution of the signal $X(t)$.

For $i = 0$, $X(t_i) = X(0)$ corresponds to the true initial distribution, and since there are no "interrupting" observations within $\mathcal{I}_0$, then the solution Eqs. (2) and (4) of Eq. (6) corresponds to the true distribution of $X(t)$ for any $t \in [t_i, t_{i+1})$. Since $u$ is constant over $\mathcal{I}_0$, then the prediction $\tilde{X}(t)$ of the Eigen-SDE solver follows Eq. (4) accurately using the analytic integration (see Appendix B; note that if $u$ were not constant, the solver would still follow the solution up to a numeric integration error). Regarding $t_1$, according to Appendix A, $\tilde{X}(t_1)$ corresponds to the true distribution of $X(t_1)$ *after* conditioning on the observation $\hat{Y}(t_1)$ (if there was an observation at $t_1$; otherwise, no filtering is needed). This completes the induction basis. Using the same arguments, if we assume for an arbitrary $i \geq 0$ that $\tilde{X}(t_i)$ corresponds to the true distribution, then $\tilde{X}(t)$ corresponds to the true distribution for any $t \in \mathcal{I}_i = [t_i, t_{i+1}]$, completing the induction.

Now, for any $t$, since $\tilde{X}(t) \sim N(\mu(t), \Sigma(t))$ is in fact the true distribution of $X(t)$, the expected square error $E[SE(t)] = E[(\mu - X(t))^2]$ is minimized by choosing $\mu := \mu(t)$; and the expected log-likelihood $E[\ell(t)] = E[\log P(X(t)|\mu, \Sigma)]$ is maximized by $\mu := \mu(t), \Sigma := \Sigma(t)$. □

## E  Extended Experiments

### E.1  Ablation Study for Patient Individualization

To provide an insight over the importance of the dynamics-individualization, we perform an ablation study for the hypernetwork module. We use the same medical benchmarks as in Section 6, and fit a version of NESDE with neutralized hypernetwork module. In particular, we fix the context inputs of the module to be a vector of 1s, and thus prevent any propagation from the context features to the model's output. The results are presented in Table 4, and show a great degradation in model

Table 4: Test mean square errors (MSE) and negative log-likelihood (NLL) in the medication-dosing benchmarks. This is an extension of Table 3 with the additional results of NESDE without the hypernetwork.

| Model | UH Dosing | | Vancomycin Dosing | |
|---|---|---|---|---|
| | MSE | NLL | MSE | NLL |
| Naive | $613.3 \pm 13.48$ | $-$ | $112.2 \pm 16.4$ | $-$ |
| LSTM | $482.1 \pm 6.52$ | $-$ | $92.89 \pm 11.3$ | $-$ |
| GRU-ODE-Bayes | $491 \pm 6.88$ | $4.52 \pm 0.008$ | $80.54 \pm 11.8$ | $6.38 \pm 0.12$ |
| CRU | $450.4 \pm 8.27$ | $4.49 \pm 0.012$ | $76.4 \pm 12.8$ | $3.87 \pm 0.2$ |
| NESDE – no hypernet | $529.7 \pm 13.34$ | $5.42 \pm 0.067$ | $87.32 \pm 11.57$ | $3.73 \pm 0.13$ |
| NESDE (ours) | $\mathbf{411.2 \pm 7.39}$ | $\mathbf{4.43 \pm 0.01}$ | $\mathbf{70.71 \pm 12.3}$ | $\mathbf{3.69 \pm 0.13}$ |

performance in the UH-dosing benchmark, approving that the hypernetwork indeed utilize the information within the context features. In the Vancomycin dosing benchmark, while we still observe a degradation comparing to NESDE, the version of NESDE without hypernetwork still outperforms LSTM in terms of MSE and the rest of the baselines (except NESDE) in terms of NLL.

### E.2 Comparison to ODE-based Methods

Section 5 compares NESDE to GRU-ODE-Bayes [De Brouwer et al., 2019] – a recent ODE-based method that can provide an uncertainty estimation (which is a typical requirement in medical applications). Similarly to other recent ODE-based methods [Chen et al., 2018], GRU-ODE-Bayes relies on a non-linear neural network model for the differential equation. GRU-ODE-Bayes presents relatively poor prediction accuracy in Section 5, which may be partially attributed to the benchmark settings. First, the benchmark required GRU-ODE-Bayes to handle a control signal. As proposed in De Brouwer et al. [2019], we incorporated the control as part of the observation space. However, such a control-observation mix raises time synchrony issues (e.g., most training input samples include only control signal without observation) and even affect the training supervision (since the new control dimension in the state space affects the loss). Second, as discussed above, the piecewise linear dynamics of NESDE provide higher sample efficiency in face of the 1000 training trajectories in Section 5.

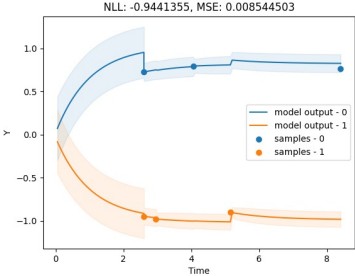

Figure 9: A sample test trajectory of the sparsely-observable OU process. The observations and the NESDE predictions (based on training over 400 trajectories) are presented separately for each of the two dimensions of the process.

In this section, we explicitly study the sample efficiency of NESDE vs. GRU-ODE-Bayes in a problem with no control signal. Specifically, we generate data from the GitHub repository of De Brouwer et al. [2019]. The data consists of irregular samples of the two-dimensional Ornstein-Uhlenbeck process, which follows the SDE

$$dx_t = \theta(\mu - x_t)dt + \sigma dWt,$$

where the noise follows a Wiener process, which is set in this experiment to have the covariance matrix

$$Cov = \begin{pmatrix} 1 & 0.5 \\ 0.5 & 1 \end{pmatrix}.$$

The process is sparsely-observed: we use a sample rate of $0.6$ (approximately 6 observations for 10 time units). Each sampled trajectory has a time support of 10 time units. The process has two dimensions, and each observation can include either of the dimensions or both of them. The dynamics

of the process are linear and remain constant for all the trajectories; however, the stable "center" of the dynamics of each trajectory (similarly to $\alpha$ in Eq. (6)) is sampled from a uniform distribution, increasing the difficulty of the task and requiring to infer $\alpha$ in an online manner.

Fig. 9 presents a sample of trajectory observations along with the corresponding predictions of the NESDE model (trained over 400 trajectories). Similarly to De Brouwer et al. [2019], the models are tested over each trajectory by observing all the measurements from times $t \leq 4$, and then predicting the process at the times of the remaining observations until the end of the trajectory.

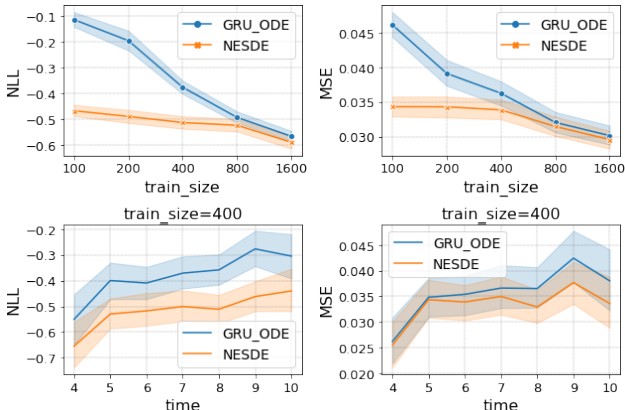

Figure 10: Top: losses of NESDE and GRU-ODE-Bayes over the OU benchmark, along with confidence intervals of 95% over the test trajectories. NESDE demonstrates higher data efficiency, as its deterioration in small training datasets is moderate in comparison to GRU-ODE-Bayes. Bottom: errors vs. time, given 400 training trajectories, where all the test predictions rely on observations from times $t \leq 4$. The advantage of NESDE becomes larger as the prediction horizon is longer.

To test for data efficiency, we train both models over training datasets with different numbers of trajectories. As shown in Fig. 10, the sparsely-observable setting with limited training data causes GRU-ODE-Bayes to falter, whereas NESDE learns robustly in this scenario. The advantage of NESDE over GRU-ODE-Bayes increases when learning from smaller datasets (Fig. 10, top), or when predicting for longer horizons (Fig. 10, bottom). This demonstrates the stability and data efficiency of the piecewise linear dynamics model of NESDE in comparison to non-linear ODE models.

### E.3 Sparse Observations

This experiment addresses the sparsity of each trajectory. We use the same benchmark as in Section 5 and generate 4 train datasets, each one contains 400 trajectories, and a test set of 1000 trajectories. In each train-set, the trajectories have the same number of data samples, which varies between datasets (4,6,8,10). The test-set contains trajectories of varying number of observations, over the same support. For each train-set, we train all the models until convergence, and test them. Fig. 11 presents the MSE over the test set, for both the complex and the real eigenvalues settings. It is noticeable that even with very sparse observations, NESDE achieves good performance. Here, GRU-ODE-Bayes appears to be more sample-efficient than CRU and LSTM, but it is less sample efficient than NESDE.

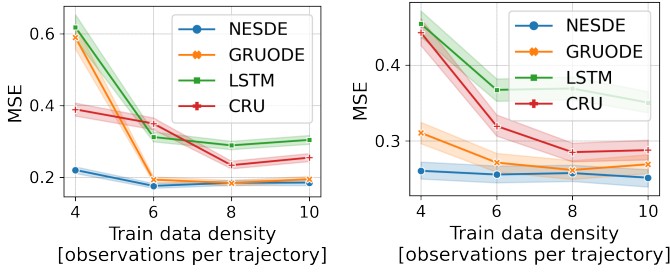

(a) Complex dynamics       (b) Real dynamics

Figure 11: Test MSE vs. train observations-per-trajectory. 95% confidence intervals are calculated over 1000 test trajectories.

### E.4   Synthetic Data Experiments with Regular Observations

While NESDE (and ODE-based models) can provide predictions at any point of time, a vanilla LSTM is limited to the predefined prediction horizon. Shorter horizons provide higher temporal resolution, but this comes with a cost: more recursive computations are needed per time interval, increasing both learning complexity and running time. For example, if medical measurements are available once per hour while predictions are required every 10 seconds, the model would have to run recursively 360 times between consecutive measurements, and would have to be trained accordingly in advance. We use the synthetic data environment from Section 5, in the *complex* dynamics setting, and test both regularly and out-of-distribution control (see Section 5). Here, we use LSTM models trained with resolutions of 1, 8 and 50 predictions per observation. All the LSTM models receive the control $u$ and the current observation $Y$ as an input, along with a boolean $b_o$ specifying observability: in absence of observation, we set $Y = 0$ and $b_o = 0$. The models consist of a linear layer on top of an LSTM layer, with 32 neurons between the two. To compare LSTMs with various resolutions, we work with regular samples, 10 samples, one at each second. The control changes in a $10^{-2}$ seconds' resolution, and contains information about the true state.

In Fig. 12c we present a sample trajectory (without the control signal) with the predictions of the various LSTMs and NESDE. It can be observed that while NESDE provides continuous, smooth predictions, the resolution of the LSTMs must be adapted for a good performance. As shown in Fig. 12a, all the methods perform well from time $t = 3$ and on, still, NESDE and the low-resolution variants of LSTM attain the best results. The poor accuracy of the high-resolution LSTM demonstrates the accuracy-vs-resolution tradeoff in recursive models, moreover, GRUODE shows similar behavior in this analysis, which may hint on the recursive components within GRUODE.

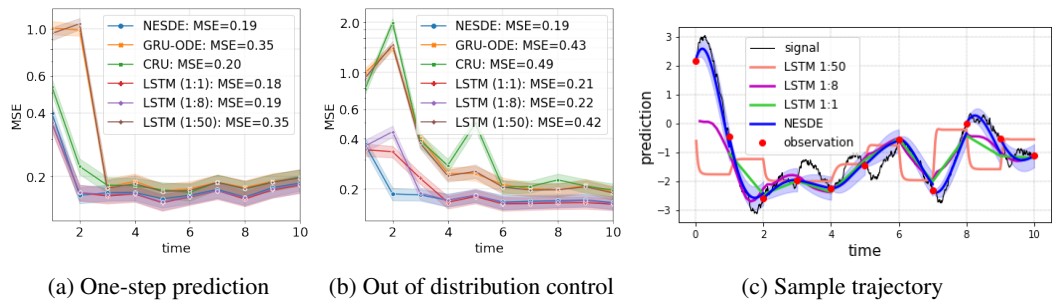

(a) One-step prediction      (b) Out of distribution control      (c) Sample trajectory

Figure 12: MSE for predictions, relying on the whole history of the trajectory for (a) the test set, and (b) out-of-distribution test set. The uncertainty corresponds to 0.95-confidence-intervals over 1000 trajectories. (c) Sample trajectory and predictions. The LSTM predictions are limited to predefined times (e.g., LSTM 1:1 only predicts at observation times), but their predictions are connected by lines for visibility. The shading corresponds to NESDE uncertainty (note that the LSTM does not provide uncertainty estimation).

The out-of-distribution test results (Fig. 12b) show that a change in the control policy could result in major errors; while NESDE achieves errors which are close to Fig. 12a, the other methods deteriorate in their performance. Notice the scale difference between the figures. The high-resolution LSTM and the ODE-based methods suffer the most, and the low-resolution variants of the LSTM, demonstrate

robustness to the control change. This result is similar to the results we present in Section 5, although here we see similarities between the variants of the LSTM and the ODE-based methods.

## E.5 Interpretability: Inspecting the Spectrum

In addition to explicit predictions at flexible times, NESDE provides direct estimation of the process dynamics, carrying significant information about the essence of the process.

For example, consider the following 3 processes, each with one observable variable and one latent variable: $A_1 = \left( \begin{smallmatrix} -0.5 & -2 \\ 2 & -1 \end{smallmatrix} \right)$ with the corresponding eigenvalues $\lambda_1 \approx -0.75 \pm 1.98i$; $A_2 = \left( \begin{smallmatrix} -0.5 & -0.5 \\ -0.5 & -1 \end{smallmatrix} \right)$ with $\lambda_2 \approx (-1.3, -0.19)^\top$; and $A_3 = \left( \begin{smallmatrix} 1 & -2 \\ 2 & -1 \end{smallmatrix} \right)$ with $\lambda_3 \approx \pm 1.71i$. As demonstrated in Fig. 4, the three processes have substantially different dynamics: roughly speaking, real negative eigenvalues correspond to decay, whereas imaginary eigenvalues correspond to periodicity.

For each process, we train NESDE over a dataset of 200 trajectories with 5-20 observations each. We set NESDE to assume an underlying dimension of $n = 2$ (i.e., one latent dimension in addition to the $m = 1$ observable variable); train it once in real mode (real eigenvalues) and once in complex mode (conjugate pairs of complex eigenvalues); and choose the model with the better NLL over the validation data. Note that instead of training twice, the required expressiveness could be obtained using $n = 4$ in complex mode (see Appendix C); however, in this section we keep $n = 2$ for the sake of spectrum interpretability.

As the processes have linear dynamics, for each of them NESDE learned to predict a consistent dynamics model: all estimated eigenvalues are similar over different trajectories, with standard deviations smaller than 0.1. The learned eigenvalues for the three processes are $\tilde{\lambda}_1 = -0.77 \pm 1.98i$; $\tilde{\lambda}_2 = (-0.7, -0.19)^\top$; and $\tilde{\lambda}_3 = -0.03 \pm 0.83i$. That is, NESDE recovers the eigenvalues class (complex, real, or imaginary), which captures the essence of the dynamics – even though it only observes one of the two dimensions of the process. The eigenvalues are not always recovered with high accuracy, possibly due to the latent dimensions making the dynamics formulation ambiguous.

## E.6 Model Expressiveness and Overfitting

It is well known that more complex models are capable to find complex connections within the data, but are also more likely to overfit the data. It is quite common that a data that involves control is biased or affected by confounding factors: a pilot may change his course of flight because he saw a storm that was off-the-radar; a physician could adapt his treatment according to some measure that is off-charts. Usually, using enough validation data could solve the overfitting issue, although sometimes the same confounding effects show in the validation data, which results in a model that is overfitted to the dataset. When targeting a model for control adjustment, it is important that it would be robust to changes in the control; a model that performs poorly when facing different control is unusable for control tuning. To exemplify an extreme case of confounding factors in the context of control, we add a correlation between the control (observed at all times) to the predictable measure (observed sparsely), in particular at times that the predictable is unobserved. We harness the same synthetic data benchmark as in Section 5, and use regular time samples, and the same LSTM baselines as in Appendix E.4 but here we generate different two types of control signals:

1. Same Distribution (SD): at each time $t$, the control $u(t) = b_t - 0.8 \cdot Y_t$.

2. Out of Distribution (OOD): at each time $t$, the control $u(t) = b_t + 0.8 \cdot Y_t$.

$b_t$ is a random piecewise constant and $Y_t$ is the exact value of the measure we wish to predict. The first type is used to generate the train and the test sets, additionally we generate an out-of-distribution test-set using the second type. We observe in Fig. 13 that GRU-ODE-Bayes and the high-resolution LSTM achieve very low MSE over the SD as seen during training. CRU also achieves very low MSE, although not as much. The results over the OOD data show that the high performance over SD came with a cost – the better a model is over SD the worse it is over OOD. The results of LSTM 1:1 are not surprising, it sees the control signal only at observation-times, so it cannot exploit the hidden information within the control signal. NESDE does not ignore such information, while maintaining the robustness w.r.t. control.

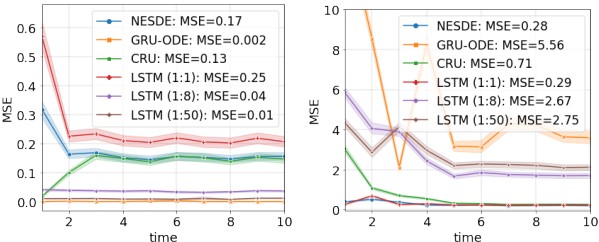

(a) Same control distribution    (b) Out of distribution control

Figure 13: MSE for predictions under regular time samples, where the control signal is correlated to the measure we wish to predict, even in times when it is unobserved. (a) Shows the results for a test set that has the same correlation between the control and the predictable measure as in the train set. (b) present the MSE for a different test set, with different correlation. Notice the different scales of the graphs.

# F    Medication Dosing Prediction: Implementation Details

Below, we elaborate on the implementation details of Section 6.

## F.1    Data preprocessing

**Heparin:** We derive our data from the MIMIC-IV dataset [Johnson et al., 2020], available under the PhysioNet Credentialed Health Data License. For the UH dosing dataset, we extract the patients that were given UH during their intensive care unit (ICU) stay. We exclude patients that were treated with discrete (not continuous) doses of UH, or with other anticoagulants; or that were tested for aPTT less than two times. The control signal (UH dosing rate) is normalized by the patient weight. Each trajectory of measurements is set to begin one hour before the first UH dose, and is split in the case of 48 hours without UH admission. This process resulted with $5866$ trajectories, containing a continuous UH signal, an irregularly-observed aPTT signal, and discretized context features. Note that we do not normalize the aPTT values.

**Vancomycin:** The VM dosing dataset derived similarly, from patients who received VM during their ICU stay, where we consider only patients with at least $2$ VM concentration measurements. Each trajectory begins at the patient's admission time, and we also split in the case of 48 hours without VM dosage. Additionally, we add an artificial observation of $0$ at time $t = 0$, as the VM concentration is $0$ before any dose was given (we do not use these observations when computing the error).

**General implementation details:** For each train trajectory, we only sample some of the observations, to enforce longer and different prediction horizons, which was found to aid the training robustness. Hyperparameters (e.g., learning rate) were chosen by trial-and-error with respect to the validation-set (separately for each model).

Context variables $C$ are used in both domains. We extract $42$ features, some measured continuously (e.g., heart rate, blood pressure), some discrete (e.g., lab tests, weight) and some static (e.g., age, background diagnoses). Each feature is averaged (after removing outliers) over a fixed time-interval of four hours, and then normalized.

## F.2    LSTM Baseline Implementation

The LSTM module we use as a baseline has been tailored specifically to the setting:

1. It includes an embedding unit for the context, which is updated whenever a context is observed, and an embedded context is stored for future use.

2. The inputs for the module include the embedded context, the previous observations, the control signal and the time difference between the current time and the next prediction time.

3. Where the control signal is piecewise constant: any time it changes we produce predictions (even though no sample is observed) that are then used as an input for the model, to model the effect of the UH more accurately.

We train it with the same methodology we use for NESDE where the training hyperparameters chosen by the best performance over the validation data.

**Architecture for the medication dosing benchmarks**: The model contains two fully connected elements: one for the context, with two hidden layers of size $32$ and $16$-dimensional output which is fed into a $Tanh$ activation; the second one uses the LSTM output to produce a one-dimensional output, which is fed into a ReLU activation to produce positive outputs, its size determined by the LSTM dimensions. The LSTM itself has an input of $19$ dimensions; $16 + 1 + 1 + 1$ for the context, control, previous observations and the time interval to predict. It has a hidden size of $64$ and two recurrent layers, with dropout of $0.2$. All the interconnections between the linear layers include ReLU activations.

**Architecture for the synthetic data benchmarks**: Here, there is no context, then the model contains one fully connected element that receives the LSTM output and has two linear layers of sizes $32$ and $1$ with a Tanh activation between them. The LSTM has an input of $3$ dimensions; for the state, control signal, and the time interval to predict. It has a hidden size of $32$ and two recurrent layers, with dropout of $0.2$.

### F.3   Extended Results

The figures below present more detailed information for the experiments discussed in Section 6.

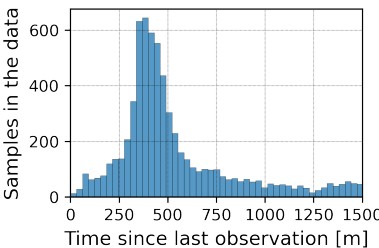

Figure 14: Histogram of prediction horizons in the UH dosing data (Section 6). Notice that the peak of the histogram around 6 hours (360 minutes) corresponds to the accuracy peak of the LSTM and GRU-ODE-Bayes in Fig. 8.

