# OpenReview forum: "Individualized Dosing Dynamics via Neural Eigen Decomposition"
_NeurIPS.cc/2023/Conference — NeurIPS 2023 poster_

### Official Review · Reviewer_a6Bv · 2023-07-05

**Soundness:** 3 good
**Presentation:** 3 good
**Contribution:** 3 good
**Rating:** 6
**Confidence:** 3

**Summary:**

This develops a robust SDE algorithm for medical forecasting. This is demonstrated on both synthetic and real data.

**Strengths:**

The problem solved is clearly compelling and the performance is evaluated on a wide range of datasets, where it exhibits superior performance to reasonable alternatives. It would appear to be an overall robust method.

**Weaknesses:**

It would be nice if some of the components of the model such as hypernet were defined here rather than briefly described, even though I understand the desire to save space. It makes it so that this paper is more difficult to understand/review.

**Questions:**

SDEs are not my area of expertise. I expect the other reviewers will have more enlightening questions and suggestions.

**Limitations:**

No societal ramifications, but no limitations mentioned as far as shortcomings of the model are concerned.

---

> ### Author Rebuttal · Authors · 2023-08-10
>
> We thank the reviewer for their comments. If our response is satisfactory, we kindly ask the reviewer to update the score.
>
> **Components Description**
>
> In the current version, we briefly describe the components in Section 4. The observation filter is further elaborated in Appendix A, the ESDE is elaborated in Appendices B, C.
> For the Hypernet, we will add to the appendices a complete presentation of the architecture, in addition to a block diagram (attached as a pdf file in the author rebuttal) that specifically explains how the Hypernet module functions within NESDE. Thanks for pointing it out.

---

> > ### Comment · Reviewer_a6Bv · 2023-08-14
> >
> > Done, my review now matches the other reviewers with more familiarity with the field.

---

### Official Review · Reviewer_hUXX · 2023-07-12

**Soundness:** 3 good
**Presentation:** 4 excellent
**Contribution:** 3 good
**Rating:** 6
**Confidence:** 3

**Summary:**

The paper presents a piece-wise linear neural SDE with control for biomedical forecasting. The key contribution is combining rigorous linear SDE solutions with hypernet parameterisation.

**Strengths:**

The paper is fantastically written and very clear.

The idea of linearising SDEs is relatively novel, and the overall model is sensible and principled.

The results are good, and show that a relatively simple model construction is perhaps ideal for medical data.

**Weaknesses:**

The method is somewhat incremental: it combines linear SDEs with hypernets.



**Questions:**

The imaginary/cyclicity argument could have been elaborated more in experiments. Is medical data cyclical? I'm not sure.

How come in fig3 we get same performance with 2 or 8 datapoints?

**Limitations:**

No issues

---

> ### Author Rebuttal · Authors · 2023-08-09
>
> We thank the reviewer for their positive feedback.
>
> **Regarding the value of the method**
>
> NESDE aims to solve a hard problem – forecasting and filtering a sparsely-observed SDE with an independent control signal. To address the accompanying challenges, NESDE combines several components. First, in this context, models that can capture complex behaviors are prone to overfitting (see the OOD experiments, Section 5). NESDE enables to inject domain knowledge and control the model’s complexity/timescale by tuning its hyperparameters; size of the latent process, type (imaginary/real/complex) of the eigenvalues, stability of the system, partition to context/state/control, and the interval length. By tuning these hyperparameters, NESDE can capture complex behavior and provide forecasts which are more robust to OOD control. Second, the ability to update future forecasts, given the current observation, is important. The filtering mechanism of NESDE adapts to new observations online, which translates to better predictions. Finally, by using the analytical solution, NESDE does not require the usage of an integrator, which is computationally expensive, and mandatory for Neural-ODE based methods.
>
>
> **The imaginary/cyclicity argument could have been elaborated more in experiments. Is medical data cyclical? I'm not sure.**
>
> The imaginary argument is a general capability of NESDE, but we did not find it useful in the dosing problems we addressed. We do include experiments with complex eigenvalues (which are cyclic and decaying) in Section 5 and in the appendix. In appendix section E.1 we show how NESDE handles pure imaginary eigenvalues.
>
> **How come in fig3 we get same performance with 2 or 8 datapoints?**
>
> The number of datapoints correspond to the filtration process: the model learns the dynamics from “offline” data, then infers “online” data in real-time. Here, there is no observation noise, then at some point (after approx’ 2 observations) the uncertainty of NESDE becomes solely the noise of the dynamics (with an additional small error) and thus cannot be greatly improved.

---

### Official Review · Reviewer_vmHv · 2023-07-12

**Soundness:** 4 excellent
**Presentation:** 4 excellent
**Contribution:** 4 excellent
**Rating:** 7
**Confidence:** 4

**Summary:**

This paper proposes a novel network approach, named Neural Eigen Stochastic Differential Equations (NESDE), to solve sequential prediction problems mainly in medical dosing control field of applications.

**Strengths:**

$\mathbf{1}.$ The proposal of NESDE is novel, along with the hypernet that determines the subsequent model parameters.

$\mathbf{2}.$ This paper is very well written. The motivation, i.e., to address challenges in sequential prediction of medical dosing control, and the limitation of current works, is very well explained. The related works and background fundamental theories are also very well stated with intuitive illustrations.

$\mathbf{3}.$ The theoretical analysis is comprehensive and sound (also seen in $\mathbf{Appendix}$.)

$\mathbf{4}.$ The empirical analysis is detailed and thorough. Author designed the experiments to validity the algorithmic merits and also real life application in the medical dosing field.

**Weaknesses:**

$\mathbf{Note: }.$ It is a unfamiliar application field to me, and thus it is hard to determine whether or not the proposed method is indeed a significantly novel approach in the field.

$\mathbf{1}.$ The major concern is whether or not the proposed method has a broader application. It seems that the direct counterpart is neural ordinary differential equations, and is it possible to discuss and compare the general applicabilities of NODEs versus NESDEs?

**Questions:**

$\mathbf{1}.$ See $\mathbf{1}.$ in $\mathbf{Weakness}$.

$\mathbf{2}.$ The experiment with synthetic data seems to support, e.g., the sample efficiency of the proposed method. However, it lacks training efficiency/computational complexity in the experiments. Is it possible to quantify the computational cost when comparing with other methods?

$\mathbf{Trivial}.$ What is the point of comparison in Figure 1?

**Limitations:**

This paper does not provide broader impact or limitation statements, but i do not have any concerns on potential negative societal impact.

---

> ### Author Rebuttal · Authors · 2023-08-09
>
> We thank the reviewer for their positive and detailed feedback.
>
> **General applicability of NESDE and Neural-ODE**
>
> NESDE constitutes a framework for forecasting individualized dosing dynamics. As described in Section 3, we consider the general problem of forecasting the dynamics of a controlled SDE. This problem is in fact a wide family of problems, including medication dosing regimes, and general controlled chemical or biological processes. Many such processes require control: fermentation in the wine-making process, irrigating and fertilizing a crop (agriculture), and monitoring chemical reactions under uncertainty. Neural-ODE on the other hand could be used for various applications (e.g., classification, regression, forecasting), when combined in different architectures.
> In the current version of our paper, in Sections 1, 1.1, we discuss the applicability of NODE-based methods to real-life problems, and in particular to the problem of forecasting dosing dynamics. In Section 5 we compare NESDE to GRU-ODE-Bayes, which is a NODE-based method for forecasting the dynamics of an SDE.
>
> Following your comment, we will add a short discussion, comparing the applicability of both methods, in Section 1.1. Thanks for bringing this to our attention.
>
> **The experiment with synthetic data seems to support, e.g., the sample efficiency of the proposed method. However, it lacks training efficiency/computational complexity in the experiments. Is it possible to quantify the computational cost when comparing with other methods?**
>
> The computational complexity could be measured in run-time, and we will add it to the appendix. Note that compared to the neural ODE methods, NESDE enjoys better run-time.
>
> **What is the point of comparison in Figure 1?**
>
> The purpose of Figure 1 is to illustrate the difficulty of the combination of sparse observations with noisy dynamics. We present two types of SDEs, not comparing them. We will clear this out in the figure's caption.

---

### Official Review · Reviewer_uGSy · 2023-07-26

**Soundness:** 3 good
**Presentation:** 3 good
**Contribution:** 2 fair
**Rating:** 6
**Confidence:** 3

**Summary:**

This paper presents a method for predicting time series at irregular time points. Its approach systematically integrates control signals, a convenient attribute for modelling dosing regimes in medicine. The latent process follows a piecewise linear stochastic differential equation (SDE) with control. The SDE is parameterized using its spectral representation, allowing for effective and analytical propagation. The parameters of the SDE are learned by a Hypernet. Experimental validation of the method is conducted on artificial data and two medication dosing problems.

**Strengths:**

- The paper introduces a method that effectively models control, with experimental results indicating an improvement over baseline models lacking explicit control.
- The well-structured experimental study underscores the applicability and value of the method for two specific medical dosing scenarios.
- The authors propose an efficient technique to compute dynamics by parameterizing the system's eigenfunction, thereby enabling fast and analytical computation.
- This parameterization strategy not only facilitates the integration of prior knowledge on the dynamics, as evidenced in the dosing experiments, but also enhances interpretability—which is a useful attribute in practise

**Weaknesses:**

- The related work sections seems comprehensive, though neural SDE approaches could also be discussed as the model is based on a SDE latent process.
- Explicitly stating which equations represent ESDE() and Filter() in Algorithm 1 would enhance clarity.
- Although the experimental study appears to be well-executed and thoughtfully designed, the paper could be enhanced by including more baseline comparisons. Specifically, a comparison with NeuralSDE models would be intriguing, as they also yield uncertainty estimates and aren't confined to the piece-wise linearity constraint.
- Minor: My understanding is that the context does not change over time, though Figure 2 suggest this on the first sight.

**Questions:**

--

**Limitations:**

The piece-wise linear assumption on the observation sequence $Y(t)$ seems a restrictive assumption and limits the type of dynamics that can be modeled with the method. (From my understanding, the model learns a piece-wise linear SDE for $X(t)$ where the observations $Y(t)$ is simply a subset of the features of $X(t)$ , i.e. $Y(t)=X(t)_{1:m}$  . Therefore, the sole non-linearity allowed in the observation sequence come from the piece-wise linear formulation.) A discussion on when the piece-wise linear assumption fails compared to other (more expressive) models is encouraged.

---

> ### Author Rebuttal · Authors · 2023-08-09
>
> We thank the reviewer for their helpful feedback. We will modify the manuscript as discussed below to address the reviewer’s comments. In particular, we will clarify the selection of baselines, which seems to be the main concern in the review. If our responses are satisfactory, we will appreciate it if the reviewer can re-evaluate our contribution accordingly.
>
> **Related Work and Baselines**
>
> As the reviewer suggests, our main baselines are indeed from the family of Neural-ODE. Specifically, we looked for Neural-ODE baselines that both handle noise and provide uncertainty estimation (so that NLL can be measured). We selected GRU-ODE-Bayes and CRU, which satisfy both properties, and have available code on GitHub. Note that despite the different ODE/SDE terminology, both methods practically implement a Neural-SDE, which must not be confused with [1] that uses noise as regularization for Neural-ODEs (without uncertainty estimates). We will clarify this in the manuscript. Thanks for pointing to it.
>
> **Explicitly stating which equations represent ESDE() and Filter() in Algorithm 1 would enhance clarity.**
>
>  We will state the equation for ESDE()  (Eq. 4). For the Filter(), we will refer to Appendix A.
>
> **Minor: My understanding is that the context does not change over time, though Figure 2 suggest this on the first sight.**
>
> The context changes over time, although in fixed time-intervals. We will add a statement that would address this unclarity.
>
> __________
>
> Regarding the limitations of the piecewise linear assumption:
> The piecewise linear assumption highly depends on the interval length (Algorithm 1, and lines 207-216), which is a hyperparameter that controls how often the (linear) dynamics change to another (linear) dynamics. As it is a hyperparameter, it could be adjusted to the specific domain -- small intervals for “highly nonlinear” domains and large intervals for domains which behave close to linear. The additional expressiveness comes at the cost of more computations, and recursive updates, which could harm the overall performance, as we see in the nonlinear models we compare to, hence the importance of adjusting it.
>
>
> __________
> [1] Liu, Xuanqing, et al. "Neural sde: Stabilizing neural ode networks with stochastic noise." arXiv preprint arXiv:1906.02355 (2019).

---

> > ### Comment · Reviewer_uGSy · 2023-08-14
> >
> > I thank the authors for responding to my comments and addressing my concerns. I have updated my rating accordingly.

---

### Author Rebuttal · Authors · 2023-08-10

We are excited that all the reviewers recommended accepting the paper, and thank them for their helpful comments. The reviewers found that the “problem solved is clearly compelling” (a6Bv), and the paper is “fantastically written and very clear” (hUXX). They describe our proposed method by “the proposal of NESDE is novel” (vmHv), “effectively models control” (uGSy), and “also enhances interpretability—which is a useful attribute in practise” (uGSy). The empirical evaluation “is detailed and thorough” (vmHv) and “underscores the applicability and value of the method” (uGSy). The results “show that a relatively simple model construction is perhaps ideal for medical data” (hUXX).

---

### Decision · Program_Chairs · 2023-09-21

**Decision:**

Accept (poster)

**Comment:**

Four reviewers evaluated the submission and scored it 7666. The agreed strengths are

* The approach (NESDE and the hypernet) is novel and interpretable
* Experimental study is carefully designed and executed, and shows good performance in two specific medical dosing scenarios

Based on the positive feedback, I recommend acceptance.